# MORTAR: EVOLVING MECHANICS FOR AUTOMATIC GAME DESIGN

## ABSTRACT

We present MORTAR, a system for autonomously evolving game mechanics for automatic game design. Game mechanics define the rules and interactions that govern gameplay, and designing them manually is a time-consuming and expert-driven process. MORTAR combines a quality-diversity algorithm with a large language model to explore a diverse set of mechanics, which are evaluated by synthesising complete games that incorporate both evolved mechanics and those drawn from an archive. The mechanics are evaluated by composing complete games through a tree search procedure, where the resulting games are evaluated by their ability to preserve a skill-based ordering over players—that is, whether stronger players consistently outperform weaker ones. We assess the mechanics based on their contribution towards the skill-based ordering score in the game. We demonstrate that MORTAR produces games that appear diverse and playable, and mechanics that contribute more towards the skill-based ordering score in the game. We perform ablation studies to assess the role of each system component and a user study to evaluate the games based on human feedback.

## 1 INTRODUCTION

Procedural content generation (PCG) is a well-studied approach in game design, concerned with the automatic creation of game content such as levels, maps, items and narratives (Shaker et al., 2016; Liu et al., 2021). PCG serves multiple purposes: enabling runtime content generation in games such as roguelikes, providing ideation tools for designers, automating the production of repetitive content, and facilitating research into creativity and design processes. Traditionally, PCG research has focused on structural aspects of games—particularly level or layout generation (Risi & Togelius, 2020)—where the goal is to produce environments that are coherent, solvable, and varied.

By contrast, comparatively little attention has been paid to the procedural generation of *game mechanics*—the underlying rules for interactions that govern gameplay. Yet mechanics play a central role in shaping the player experience, determining not just how players act, but what kinds of strategies and emergent behaviours are possible. Designing mechanics is inherently challenging: unlike levels, which can be evaluated by solvability or novelty, the utility of a mechanic depends on the dynamics it induces within the context of a game. This makes both generation and evaluation significantly harder.

A central premise of this work is that evaluating game mechanics is fundamentally more difficult than evaluating assets or level layouts. Unlike these forms of content, a mechanic *cannot be judged in isolation*—it only gains meaning through the gameplay it enables. A mechanic that appears novel or complex may still be uninteresting if it does not support skill-based interaction. This insight motivates our approach: effective automation of mechanic design requires not only a generative model, but also a principled way to assess a mechanic's utility in the context of play.

We address this challenge by introducing a mechanic-centric framework for automatic game design. The central idea is to evolve mechanics not in isolation, but through their contribution to the quality of full games. Specifically, we evaluate mechanics by constructing complete games around them, and measuring whether the resulting games induce a skill-based ordering over players of different capabilities. This allows us to define a concrete notion of usefulness for a mechanic: its contribution to the overall expressivity and skill gradient of the games in which it appears.

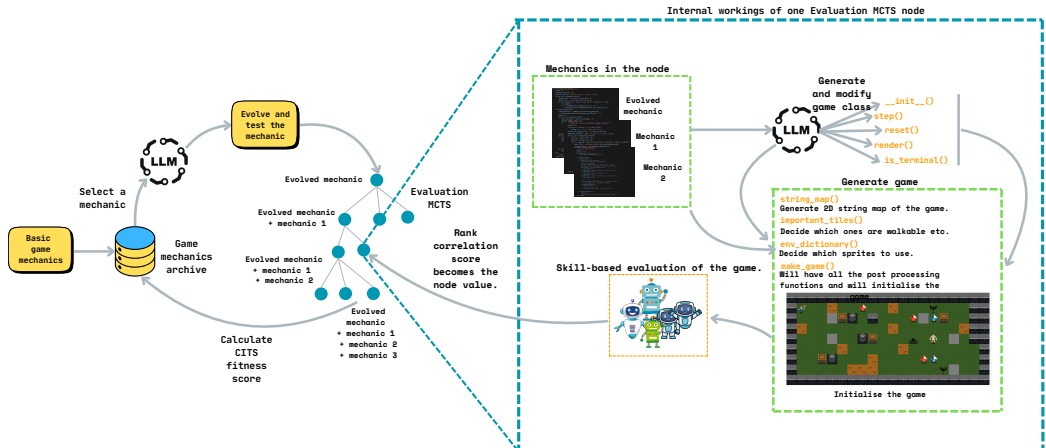

Figure 1: A flow diagram of MORTAR

We introduce MORTAR, a system that evolves game mechanics using a quality-diversity algorithm guided by a large language model (LLM). MORTAR maintains a diverse archive of mechanics, represented as code snippets, which are mutated and recombined through LLM-driven evolutionary operators. Each evolved mechanic is evaluated by embedding it into full games constructed via Monte Carlo Tree Search, which incrementally builds games by composing mechanics from the archive. These games are evaluated based on their ability to induce a consistent skill-based ranking over a fixed set of agents. We define a novel fitness measure, which quantifies the contribution of a mechanic to the final game's skill-based ordering, inspired by Shapley values (Shapley, 2016).

We demonstrate that MORTAR can evolve a diverse set of game mechanics that contribute to the quality and playability of the generated games.[1] The resulting games exhibit coherent structure, varied interaction patterns, and meaningful skill gradients. Through ablations, we show that both the tree-search-based composition and the LLM-driven mutations are critical for generating high-quality mechanics. Our results highlight the potential for using LLMs not only as generators, but as evaluators and collaborators in the game design loop.

The system described here is a research prototype for the purposes of understanding how to best generate complementary game mechanics. However, it could also serve as an ideation tool for game designers, suggesting new mechanics and mechanic combinations, perhaps in response to designer input. It is not meant to generate complete games, and aims to empower rather than replace game designers.

## 2 METHOD

MORTAR is an evolutionary algorithm for generating game mechanics, where a large language model (LLM) is used to implement code-level variation operators. A core principle of the method is that a mechanic's value lies in the gameplay it enables; mechanics are evaluated not in isolation, but by the contribution they make to full games. We formalise this through a notion of *importance*, which guides the search process.

### 2.1 EVOLUTION SETUP

MORTAR employs a Quality-Diversity (QD) algorithm, using a fixed 2D archive (as in MAP-Elites (Mouret & Clune, 2015)) to store and explore diverse game mechanics. We refer to this structure as the *Mechanics Archive*. Each mechanic is represented as a Python function belonging to a game class, and placed in the archive based on two behavioural descriptors:

---

[1]Play the generated games at: https://mortar-x3p7.onrender.com/

1. **Mechanic Type**: A categorical descriptor indicating the gameplay behaviour the mechanic enables. We define 8 mechanic types (detailed in Section 3), each associated with 10 descriptive category words. To classify an evolved mechanic, we compute similarity scores between the mechanic's name and all category words, creating a normalised similarity vector. The mechanic type is determined by identifying the highest similarity score's index and multiplying this index by the score to produce a positional similarity value that serves as the behavioural descriptor.

2. **Code Complexity**: Computed using weighted Abstract Syntax Tree (AST) analysis. We parse the mechanic's code into an AST representation and calculate complexity as a weighted sum of function calls, assignments, and return statements. Function calls receive the highest weight, as mechanisms requiring more function calls exhibit greater complexity. Assignments are weighted to reflect that additional variables may enable more interesting behaviours. Return statements contribute to complexity scoring because multiple exit paths can produce diverse behavioural outcomes.

Mechanics are selected from the archive and modified using several LLM-implemented evolutionary operators: *Mutation* adds new functionality to a single mechanic; *diversity mutation* samples three mechanics and prompts the LLM to generate a behaviorally distinct variant; *crossover* merges two mechanics (selected based on AST similarity) into a functional combination that integrates elements from both; and *compatibility mutation* generates mechanics that complement existing ones in a game, primarily used during game evaluation (see subsequent sections).

## 2.2 Evaluating Game Mechanics

Each evolved mechanic is represented as a function within a Python class. To prepare it for evaluation, we prompt the LLM to construct the rest of the class around it in a step-by-step fashion, starting with the `__init__()` method to define any required variables and scaffolding, `step` method to add actions, `reset` and `render` method as needed.

The mechanic is then tested for syntax and runtime errors. If no errors occur, we simulate it in a static test environment with simple objects and characters for the mechanic to interact with them, if necessary. A Monte Carlo Tree Search (MCTS) agent is used to interact with the environment, verifying that the mechanic is functional and non-trivial. Only mechanics that pass both tests proceed to the *usefulness* evaluation stage. Failed mechanics are discarded to reduce unnecessary LLM calls.

## 2.3 Automated Game Construction

To evaluate a mechanic's usefulness, we embed it within a full game. Games are constructed through MCTS, where the root node is the evolved mechanic, and each expansion adds a new mechanic that is either sampled from the archive or generated via compatibility mutation. Each path through the tree represents a particular combination of mechanics; that is, a complete game.

Games are also implemented as Python classes, following a common template with core methods, such as `step`, `reset`, `render`, move mechanics and preset variables. The LLM is prompted to modify or add functionality to these methods as needed, in an iterative manner. It is also asked to define any helper methods or variables required by the mechanics. At the end of this process, the LLM is prompted to define a win condition and generates a corresponding termination function. It also selects appropriate tiles from a predefined set, maps them to characters, and generates a 2D string-based level layout using these mappings. A final function defines which tiles are walkable, interactive, or character-specific.

The complete game script includes the game class, the tile and map generation functions, and a preset function that instantiates the full game. Simple postprocessing ensures the map is rectangular, contains exactly one player, and is free of formatting issues (e.g. whitespace padding).

## 2.4 Evaluation of the Game

A central idea in our work is that a game's quality is revealed through the emergence of a consistent skill gradient or game depth: a well-designed game should allow players of differing abilities to be meaningfully distinguished. We implement this by evaluating how well the game ranks a fixed set

of players by skill. This approach allows us to evaluate not just whether a game is playable, but whether it rewards skill—a more robust signal of design quality.

To assess whether a game rewards skill, we fix a pool of five agents with varying ability levels, inspired by Nielsen et al. (2015). These include three MCTS agents with increasing numbers of rollouts, a random agent, and an agent that takes no actions.[2] This defines a clear expected skill ordering: the strongest agent should be the MCTS variant with the most rollouts, followed by the medium and low rollout agents, then the random agent, and finally the no-op agent. The outcome rank is induced by playing the game and recording empirical win rates. To quantify alignment between the expected and outcome rankings, we compute Kendall's Tau ($\tau$), a standard measure of rank correlation: $\tau = \frac{C-D}{\frac{p(p-1)}{2}}$. Here, $C$ and $D$ are the number of concordant and discordant pairs, respectively, and $p$ is the number of players (five in this case). Concordance occurs when the relative ranking between two players agrees between the expected and observed orders; discordance occurs when they disagree. A value of 1 indicates perfect alignment with the expected ranking, 0 indicates no correlation, and $-1$ reflects a completely reversed order. We consider a game unplayable if $\tau = -1$.

While $\tau$ provides a global measure of game quality, it reveals nothing about the source of that quality. To address this, we introduce *Constrained Importance Through Search* (CITS), a scoring method to measure each mechanic's marginal contribution to the emergence of a skill gradient. Inspired by Shapley values Shapley (2016), CITS estimates how much each mechanic contributed to the final game's $\tau$ score. However, computing full Shapley values would require evaluating every subset of mechanics–exponential in the number of mechanics. Instead, CITS is defined over the exploration tree constructed during generation, making it computationally tractable and grounded in actual gameplay evaluations. Formally, the CITS score for mechanic $i$ is:

$$\text{CITS}_i = \frac{1}{|N_i|} \sum_{n \in N_i} \phi_i^{(n)},$$

where $N_i = \{n \in T : i \in M_n, n \neq n_{root}\}$ is the set of non-root nodes in the tree $T$ that contain mechanic $i$, and $M_n$ is the mechanic set at node $n$. The contribution $\phi_i^{(n)}$ is computed using the standard Shapley formula over the restricted set of explored subsets:

$$\phi_i^{(n)} = \sum_{S \subseteq M_n \setminus \{i\}} \frac{|S|! \cdot (|M_n| - |S| - 1)!}{|M_n|!} \cdot \Delta_i^{(n)}(S),$$

where the marginal value term is defined as the difference in value when adding mechanic $i$ to the subset $S$:

$$\Delta_i^{(n)}(S) = v_T(S \cup \{i\}) - v_T(S).$$

Finally, the value function $v_T(S)$ returns the $\tau_m$ score for the node $m$ with exactly mechanics $S$, if such a node exists in the tree; otherwise, it is defined to be 0:

$$v_T(S) = \begin{cases} \tau_m & \text{if } \exists m \in T \text{ s.t. } M_m = S \\ 0 & \text{otherwise} \end{cases}$$

This search-constrained Shapley approach allows us to assess a mechanic's value in context, measuring its contribution within actual, discovered game designs rather than hypothetical combinations. As such, the CITS score provides a principled, interpretable, and efficient mechanism for attributing gameplay quality to individual mechanics.

---

[2]Any agents with a clear capability ordering would suffice, such as heuristic agents with different search depths, or reinforcement learning agents with varying training budgets.

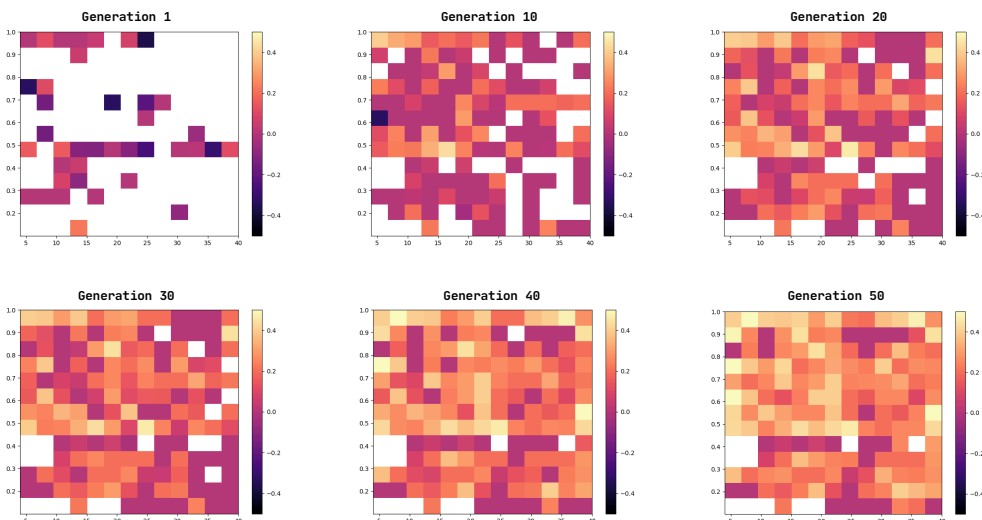

Figure 2: Coverage of game mechanic archive over a run.

## 3 EXPERIMENT SETUP

MORTAR employs a 2D Quality-Diversity (QD) archive with dimensions for mechanic type (0–1.0) and code complexity (4–40), forming a $13 \times 13$ grid. The first dimension categorises mechanics into nine types: *movement, interaction, combat, progression, environment, puzzle, resource management, exploration, time manipulation*. To categorise a mechanic, we use DistilBERT (Sanh et al., 2019) embeddings to compute the similarity between mechanic function names and associated category words (detailed in the Appendix C). The complexity dimension and archive ranges were determined through experimentation to maximise archive coverage.

The system operates with a batch size of 10, selecting individuals from the archive and applying evolutionary operators in parallel. Operator selection probabilities are 50% for diversity mutation, 30% for mutation, and 20% for crossover. Diversity mutation samples three mechanics, while crossover selects pairs based on AST similarity. The static environment used to evaluate the evolved mechanic in isolation can be found in the Appendix A.

For game construction, we use MCTS with 20 iterations, where each expansion adds one mechanic per node (maximum 3 children per node). Unlike traditional MCTS, we do not simulate; instead, we evaluate the complete game formed by all mechanics on the path from root to the newly expanded node, then backpropagate before proceeding to the next expansion. Compatibility mutation generates new mechanics within nodes, with a 50% probability of creating novel mechanics versus selecting from the existing archive. All LLM operations use GPT-4o-mini for both evolution and game creation. Skill assessment employs five agents with a clear capability ordering: MCTS variants with 100,000, 10,000, and 1,000 iterations, plus random and no-action agents for Kendall's Tau rank correlation computation.

We conduct extensive ablation studies, replacing the MCTS procedure with three alternatives: random mechanic selection, LLM-prompted selection, and greedy fitness-based selection. Each method generates games with 1-4 mechanics to compute CITS scores. We then conduct another ablation with a *Sokoban* (Murase et al., 1996) level as the initial game. All experiments are averaged over five runs due to computational constraints (approximately \$30–50 per run with GPT-4o-mini).

Our evaluation metrics assess MORTAR's progression through multiple measures. The Quality-Diversity (QD) Score sums all fitness values to indicate improving mechanic quality. Accumulated Rank Correlation totals Kendall's $\tau$ scores across all tree nodes. We track both maximum and mean fitness scores via CITS evaluation, monitor the number of elites filling the archive, and calculate game creation success rate as the proportion of functional games among all generated games.

Furthermore, a user study is conducted to get feedback to known if the games are actually interesting or not. We provide 6 generated games, and pair them together according to their distribution. We then ask the user to play the games and mark which one of the two is more *interesting*, *novel*, *fun to play*, and *easy to understand*. We also give them an option of *Neither*, which is very important to us as it will let us know if the games are actually meaningful.

## 4 RESULTS

In this section, we analyse results from the complete MORTAR pipeline, ablation and user studies. The QD score, which sums fitnesses of all archive individuals, demonstrates MORTAR's ability to evolve increasingly better mechanics over time (Figure 3a). Figure 3b reveals complementary patterns: mean fitness (CITS score) increases gradually across generations while maximum fitness shows stepwise improvements, indicating MORTAR's capacity for continued mechanic discovery. Figure 3c provides additional evidence of progression through the accumulative Kendall's $\tau$ rank correlation score per Evaluation MCTS tree, showing that MORTAR increasingly identifies engaging games that exhibit meaningful skill-based player rankings across generations.

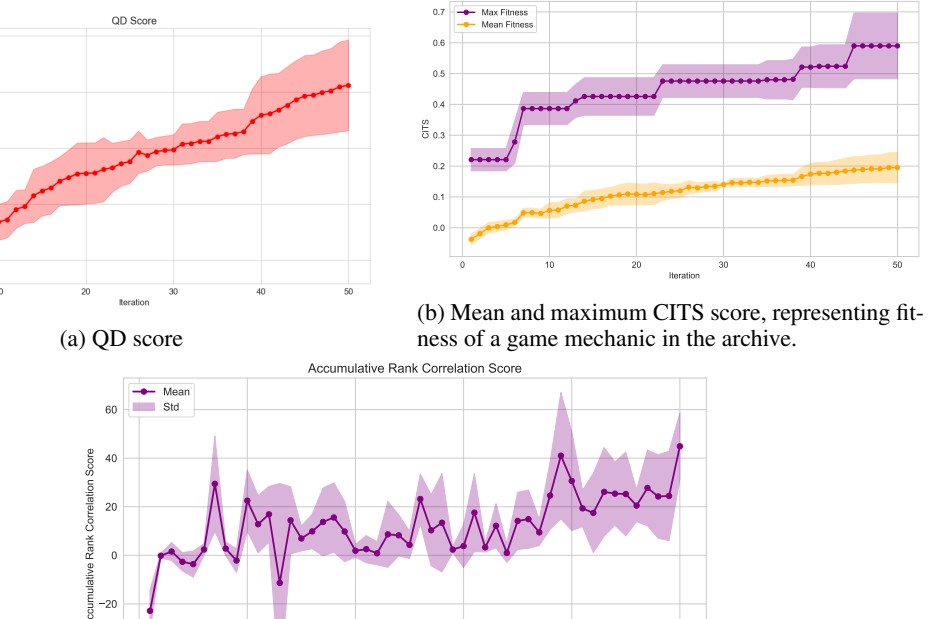

(a) QD score

(b) Mean and maximum CITS score, representing fitness of a game mechanic in the archive.

(c) Accumulative Kendall's $\tau$ rank correlation score.

Figure 3: Performance metrics over evolutionary generations.

Table 1 compares MORTAR with alternative approaches to the MCTS evaluation component, our core methodological contribution. MORTAR demonstrates superior evolvability through the highest archive coverage and consistently achieves the best QD score, maximum CITS score, and mean CITS score, indicating its ability to discover higher-quality mechanics. While Greedy Search achieves a marginally better game creation success rate—likely because it always selects the most fit mechanics—this suggests that highly fit mechanics have greater potential for generating playable games. However, MORTAR's comprehensive performance across multiple metrics demonstrates the effectiveness of its tree search-based composition approach for mechanic evolution. Furthermore, *Sokoban Initialisation* suggests that the MORTAR is sensitive to the initial mechanics and game layout, which impacts evolvability, as evidenced by the very low number of elites in this case.

Table 1: Comparison of MORTAR with alternative mechanic selection strategies: LLM-based selection, random selection, and greedy fitness-based selection across quality-diversity metrics.

| Method | No. of elites ↑ | QD score ↑ | Max CITS ↑ | Mean CITS ↑ | Games success rate ↑ |
|---|---|---|---|---|---|
| Evaluation MCTS (ours) | **155 ± 4.51** | **31.18 ± 8.10** | **0.59 ± 0.11** | **0.20 ± 0.05** | 16.97 ± 4.64 |
| LLM Selection | 141 ± 5.83 | 17.64 ± 4.91 | 0.27 ± 0.09 | 0.13 ± 0.06 | 11.69 ± 5.14 |
| Random Selection | 144 ± 9.10 | 9.86 ± 6.71 | 0.14 ± 0.08 | 0.06 ± 0.04 | 11.77 ± 4.19 |
| Greedy Selection | 139 ± 5.15 | 25.37 ± 2.81 | 0.51 ± 0.06 | 0.18 ± 0.13 | **18.24 ± 1.19** |
| Sokoban Initialisation | 110 ± 10.52 | 15.19 ± 3.12 | 0.45 ± 0.12 | 0.19 ± 0.07 | 15.11 ± 2.83 |

Figures 4 and 5 showcase two games generated by MORTAR, demonstrating diversity in level layouts, win conditions, and mechanics. *AllyCraft* (Figure 4) presents a challenging strategic experience where players control both their character and summoned allies, with escalating difficulty requiring versatile tactics. Effective strategies involve summoning allies strategically and eliminating enemies in optimal sequences. This game achieves a Kendall's $\tau$ of 0.8, maintaining clear agent rankings despite low overall rewards, with only minor rank switching between the do-nothing and random agents due to negative scoring.

By contrast, *TreasureHunt* (Figure 5) exhibits a Kendall's $\tau$ of 0.4, showing significant rank distortion except for the top-performing agent. This lower correlation suggests reduced strategic depth—once players discover the optimal path, the game loses replay value. *AllyCraft*'s higher $\tau$ score correlates with sustained engagement through multiple viable strategies, while *TreasureHunt*'s deterministic solution path limits long-term interest. Both games incorporate sophisticated mechanics, including ally summoning, multi-unit control, and pathfinding algorithms. The complete evolved code for these mechanics is provided in Appendix B.

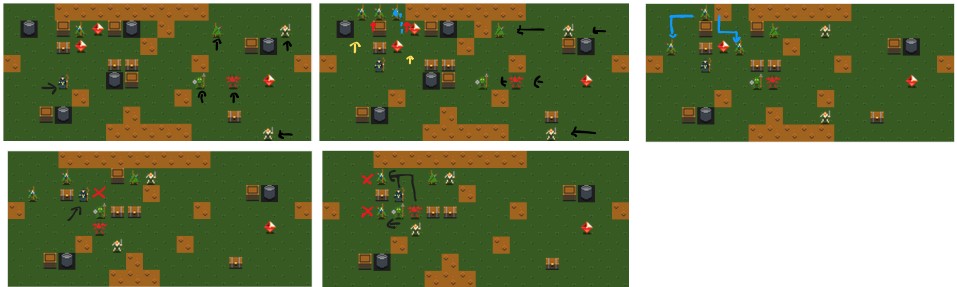

Figure 4: *AllyCraft* gameplay sequence: (Top left) Initial state showing black-marked enemies to defeat and items to collect for rewards. (Top centre) Player spawns and controls allies as additional units. (Top right) Allies collect items while enemies advance each turn. (Bottom left) One ally is defeated by an enemy while simultaneously eliminating an opposing unit. (Bottom centre) Player and remaining ally attempt coordinated attack but are overwhelmed by enemies, resulting in a loss.

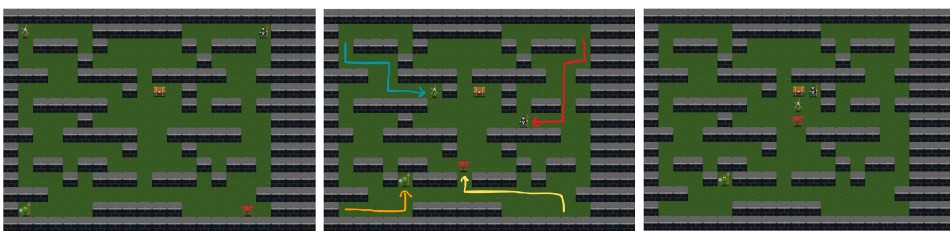

Figure 5: *TreasureHunt* gameplay sequence: (Left) Initial game state showing treasure objective in a capture-the-flag style layout. (Centre) Player spawns at the top-left corner (blue marker) while enemies begins pursuit. (Right) Final state showing close competition between player and red-marked enemy, with victory determined by action processing order. The game features an evolved A* pathfinding algorithm for enemy movement (code in Appendix B).

## 4.1 USER STUDY

To determine whether the quantitative metrics correlate with human preferences, we conducted a small comparative user study with 10 participants who evaluated 6 games generated by MORTAR across five dimensions: interestingness, novelty, frustration level, fun factor, and ease of understanding. Table 2 presents the results alongside each game's Kendall's $\tau$ score for comparison with MORTAR's automated evaluation.

The study compared three pairs of games: *TreasureHunt* versus *HuntBreakout* (capture-the-flag variants where *HuntBreakout* adds wall-breaking mechanics), *AllyCraft* versus *CrystalCavernsCommander* (RPG-style games differing in ally control mechanisms), and *MagneticProwess* versus *HeroHunt* (Sokoban-based games with magnetic pulling and enemy combat mechanics, respectively). See Appendix D for games in the user study.

We observe a general correlation between the total human preference score and MORTAR's calculated Kendall's $\tau$ values. In the first comparison, the $\tau$ difference has a smaller magnitude than the total score difference, yet both favour the same game. The second comparison shows alignment in both magnitude and direction between total score and $\tau$. However, the third comparison reveals opposing trends where the total score contradicts $\tau$, though this discrepancy may reflect the inherent difficulty of aligning automated skill-based metrics with subjective human preferences across diverse game genres.

Treating the total score as a meaningfulness metric—comprising interestingness, novelty, fun factor, ease of understanding, minus frustration—the "Neither" votes provide additional insight. These scores (1, 7, and 2 across the three comparisons, respectively) indicate that games in the second comparison are perceived as less meaningful, likely due to excessive complexity. This finding aligns with intuitive game design principles: mini-games benefit from appropriate rather than maximal complexity. While complexity can enhance engagement in full games through progressive difficulty scaling, these mini-game experiences demonstrate reduced effectiveness when sophisticated mechanics overwhelm fundamental gameplay elements. Finally, qualitative participant feedback consistently highlighted visual limitations, particularly the absence of animations and restricted sprite sets—a known limitation of MORTAR's current implementation.

Table 2: User study results comparing games. Values indicate the number of participants (out of 10) selecting each option. Total score represents the sum of positive metrics minus "Frustrating".

| Games | Interesting ↑ | Novel ↑ | Frustrating ↓ | Fun to play ↑ | Easy to understand ↑ | Total ↑ | $\tau$ ↑ |
|---|---|---|---|---|---|---|---|
| *TreasureHunt* | 0 | 1 | 3 | 1 | 4 | 3 | 0.4 |
| *HuntBreakout* | 8 | 8 | 5 | 7 | 4 | 22 | 0.5 |
| Both | 1 | 0 | 0 | 1 | 2 | 2 | — |
| Neither | 1 | 1 | 2 | 1 | 0 | 1 | — |
| *AllyCraft* | 7 | 6 | 5 | 6 | 3 | 17 | 0.8 |
| *CrystalCavernsCommander* | 2 | 2 | 3 | 3 | 2 | 6 | 0.3 |
| Both | 0 | 1 | 2 | 0 | 1 | 0 | — |
| Neither | 1 | 1 | 0 | 1 | 4 | 7 | — |
| *MagneticProwess* | 4 | 4 | 5 | 4 | 3 | 10 | 0.6 |
| *HeroHunt* | 5 | 5 | 2 | 5 | 3 | 16 | 0.3 |
| Both | 0 | 0 | 1 | 0 | 3 | 2 | — |
| Neither | 1 | 1 | 2 | 1 | 1 | 2 | — |

## 5 RELATED WORK

The core focus of MORTAR is evolving game mechanics to serve as an ideation and prototyping tool for game designers and generate novel games for testing learning algorithms. This research falls under Automatic Game Design (AGD), pioneered by (Nelson & Mateas, 2007), who formalized game mechanics through WordNet to generate micro games. Browne (2008) and Togelius & Schmidhuber (2008) independently proposed evolutionary approaches to AGD across different domains. The latter introduced learnability as a quality criterion, inspiring various approximations of skill differentiation over the years (Nielsen et al., 2015; Khalifa et al., 2017) that influence our current approach. Related concepts include game depth (Lantz et al., 2017) and formalisms for measuring a game's ability to distinguish among agents (Stephenson et al., 2020).

Non-evolutionary AGD approaches include constraint solvers for mechanics generation (Zook & Riedl, 2014) and autoencoders for learning and generating mechanics (Rieder, 2018). Recent work incorporates LLMs into AGD pipelines: ScriptDoctor generates PuzzleScript games (Earle et al., 2025), while Gavel evolves Ludii games using LLMs and Quality-Diversity algorithms (Todd et al., 2024). Similar approaches have generated 2-player games using XML-based languages (Jorge & Antonio J, 2023). MORTAR distinguishes itself by leveraging the full expressiveness of Python code generation, creating a search space that scales with advancing LLM capabilities.

MORTAR also relates to LLM-driven Procedural Content Generation (Togelius et al., 2011; Shaker et al., 2016; Liu et al., 2021). Early work included Sokoban level generation using GPT-2/3 (Todd et al., 2023), MarioGPT for Super Mario Bros levels with Novelty Search (Sudhakaran et al., 2023), and human-in-the-loop GPT-3 fine-tuning (Nasir & Togelius, 2023). Word2World and Word2Minecraft generate 2D and 3D games with fixed mechanics (Nasir et al., 2024a; Huang, 2025). MORTAR extends this paradigm by generating multiple game aspects, including mechanics and levels.

Finally, MORTAR contributes to research on LLMs as evolutionary operators in Quality-Diversity algorithms like MAP-Elites (Mouret & Clune, 2015). This approach has been applied to robot morphology evolution (Lehman et al., 2023), neural architecture search using CVT-MAP-Elites (Nasir et al., 2024b), and Ludii game generation (Todd et al., 2024).

## 6 LIMITATIONS

While MORTAR successfully generates novel game mechanics and coherent games with semantically meaningful CITS scores, several limitations warrant future investigation. The system currently modifies game rendering functions without incorporating animations, limiting visual richness. Our experiments used a relatively modest LLM (GPT-4o-mini); stronger models could potentially yield more sophisticated mechanics and improved code quality. The current 2D top-down perspective constrains the search space—extending to 3D environments would significantly expand creative possibilities.

Archive initialisation presents another challenge, as improved seeding strategies could enhance convergence and final quality. Similarly, increasing MCTS iterations during evaluation might produce higher-quality games at the cost of computational resources. Perhaps most significantly, MORTAR's autonomous evolution lacks designer control mechanisms. A controllable variant that accepts design constraints or preferences could better serve as an ideation tool, allowing game developers to guide the search toward specific gameplay goals while maintaining the system's creative discovery capabilities.

## 7 CONCLUSION AND FUTURE DIRECTIONS

We present MORTAR, a novel system for generating games through mechanic evolution. MORTAR combines MAP-Elites, a Quality-Diversity algorithm, with LLM-driven code-level mechanic evolution. The system evaluates mechanics through MCTS, which constructs complete games in each tree node and assesses them using skill-based ranking. We introduce the Constrained Importance Through Search (CITS) score, derived from Shapley values, which quantifies a mechanic's contribution within the actually searched combination space rather than hypothetical alternatives.

Our quantitative results demonstrate MORTAR's high evolvability and progressive improvement across generations through comprehensive ablation studies. Qualitative analysis reveals that games with higher scores exhibit greater strategic depth and complexity, while MORTAR consistently produces diverse gaming experiences with sophisticated mechanic interactions.

MORTAR offers several promising research directions. As an ideation tool, it could support game designers by suggesting novel mechanic combinations responsive to design constraints. The system's scalability suggests that initialisation with extensive mechanic libraries and extended evolution periods could explore previously undiscovered regions of game design space. The generated games provide rich environments for testing generalisation in reinforcement learning agents (Sutton et al., 1999), offering diverse challenges with measurable skill gradients.

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

## A PROMPTS

### A.1 GAME MECHANIC GENERATION PROMPTS

Prompts provided to MORTAR are accompanied by predefined Python code, which is then modified as required. The following methods of a class are defined separately so they can be invoked in individual prompts:

```python
init_method = r"""def __init__(self, walkable_tiles,tiles_without_char,
    tiles, str_map_without_chars, str_map, interactive_object_tiles,
    enemy_tiles, render_mode="human"):
    super(GameMechEnv, self).__init__()
    self.map_str_without_chars = str_map_without_chars.strip().split('\n
    ')
    self.map_str = str_map.strip().split('\n')
    self.map = [list(row) for row in self.map_str]
    self.map_without_chars = [list(row) for row in self.
    map_str_without_chars]
    self.tiles = tiles
    self.tiles_without_char = tiles_without_char
    self.action_space = spaces.Discrete(self.get_action_space())
    self.char_set = {'A': 0, 'B': 1, 'C': 2, 'D': 3, 'O': 4, '@': 5, '#':
     6, '&': 7}
    self.char_to_int = lambda c: self.char_set.get(c, 0)
    self.mechanic_to_action = self.get_mechanics_to_action()
    self.done = False
    self.tile_size = 16
    self.char_tile_size = 16
```

```python
        self.frames = []
        max_width = max(len(row) for row in self.map_str)
        self.observation_space = spaces.Box(
            low=0,
            high=1,
            shape=(len(self.char_set), len(self.map_str), max_width),  # Use
    len(self.char_set) for channels
            dtype=np.int32
        )

        self.default_walkable_tile = 'A'
        self.render_mode = "rgb_array"
        self.walkable_tiles = walkable_tiles
        self.interactive_object_tiles = interactive_object_tiles
        self.enemy_tiles = enemy_tiles
        self.picked_objects = []
        self.npc_tiles = ["&"]
        self.enemy_tiles = ["#"]
        self.player_health = 100
        self.enemy_health = 100
        self.current_score = 0
        self.map = [list(row) for row in self.map_str]
        self.grid_width = max(len(row) for row in self.map)
        self.grid_height = len(self.map)
        for i, row in enumerate(self.map):
            for j, tile in enumerate(row):
                if tile == '@':
                    self.player_position = (i, j)

        self.reset()"""

        reset_method = """def reset(self, seed=None):
        self.map = [list(row) for row in self.map_str]
        self.map_without_chars = [list(row) for row in self.
    map_str_without_chars]
        self.grid_width = max(len(row) for row in self.map)
        self.grid_height = len(self.map)
        for i, row in enumerate(self.map):
            for j, tile in enumerate(row):
                if tile == '@':
                    self.player_position = (i, j)
        self.current_tile = self.map_without_chars[self.player_position[0]][
    self.player_position[1]]  # Set current tile to the player's starting
     position
        return self.get_state()["map"]"""

        render_method = """def render(self, mode='human'):

        env_img = Image.new('RGBA', (len(self.map[0]) * self.tile_size, len(
    self.map) * self.tile_size))

        # 1st layer: Default walkable tile
        for i in range(len(self.map)):
            for j in range(len(self.map[0])):
                tile_img = self.tiles[self.default_walkable_tile].resize((
    self.tile_size, self.tile_size))
                env_img.paste(tile_img, (j * self.tile_size, i * self.
    tile_size), tile_img)

        # 2nd layer: Map without characters
        for i, row in enumerate(self.map_without_chars):
            for j, tile in enumerate(row):
                if tile in self.tiles and tile != self.default_walkable_tile:
                    tile_img = self.tiles[tile].resize((self.tile_size, self.
    tile_size))
```

```python
                    env_img.paste(tile_img, (j * self.tile_size, i * self.
        tile_size), tile_img)

        # 3rd layer: Characters and objects
        for i, row in enumerate(self.map):
            for j, tile in enumerate(row):
                if tile in self.tiles and tile not in self.walkable_tiles:
                    if tile.isalpha():
                        tile_img = self.tiles[tile].resize((self.tile_size,
        self.tile_size))
                    else:
                        tile_img = self.tiles[tile].resize((self.
        char_tile_size, self.char_tile_size))
                        # Center the character in the tile
                        x_offset = (self.tile_size - self.char_tile_size) //
        2
                        y_offset = (self.tile_size - self.char_tile_size) //
        2
                        env_img.paste(tile_img, (j * self.tile_size +
        x_offset, i * self.tile_size + y_offset), tile_img)

        frame = np.array(env_img.convert('RGB'))
        self.frames.append(frame)
        return frame"""

    get_action_space_method = """def get_action_space(self):
        return 3"""

    get_mechanics_to_action_method = """def get_mechanics_to_action(self)
        :
        return {
            "move_player": 0,  # 0-3 for movement
        }"""

    move_player = """def move_player(self, action):
        moves = {0: (-1, 0), 1: (1, 0), 2: (0, -1), 3: (0, 1)}  # Up, Down,
        Left, Right
        dx, dy = moves[action]
        new_row = self.player_position[0] + dx
        new_col = self.player_position[1] + dy
        reward = 0
        if 0 <= new_row < len(self.map) and 0 <= new_col < len(self.map[0]):
            new_tile = self.map[new_row][new_col]
            if new_tile in self.walkable_tiles:
                self.update_player_position(new_row, new_col, new_tile)
        return reward"""

    other_methods = r"""def update_player_position(self, new_row, new_col
        , new_tile):
        # Validate both current and new positions are within bounds
        if not (0 <= new_row < self.grid_height and 0 <= new_col < self.
        grid_width):
            return

        if not (0 <= self.player_position[0] < self.grid_height and 0 <= self
        .player_position[1] < self.grid_width):
            # If current position is invalid, just set the new position
            self.player_position = (new_row, new_col)
            self.current_tile = new_tile
            self.map[new_row][new_col] = '@'
            return

        if new_tile not in self.walkable_tiles:
            return
```

```
128        # Reset the player's previous position to the original tile
129        self.map[self.player_position[0]][self.player_position[1]] = self.
           current_tile
130        self.map_without_chars[self.player_position[0]][self.player_position
           [1]] = self.current_tile
131
132        # Update the player's position
133        self.player_position = (new_row, new_col)
134        self.current_tile = new_tile
135        self.map[new_row][new_col] = '@'
136
137 def find_player_position(self):
138        for i, row in enumerate(self.map):
139            for j, tile in enumerate(row):
140                if tile == '@':
141                    return (i, j)
142        return None
143
144 def clone(self):
145
146        new_env = GameMechEnv(
147            walkable_tiles=self.walkable_tiles,
148            tiles_without_char=self.tiles_without_char,
149            tiles=self.tiles,
150            str_map_without_chars='\n'.join(self.map_str_without_chars),
151            str_map='\n'.join(self.map_str),
152            interactive_object_tiles=self.interactive_object_tiles,
153            enemy_tiles=self.enemy_tiles
154        )
155        new_env.map = [row[:] for row in self.map]
156        new_env.map_without_chars = [row[:] for row in self.map_without_chars
           ]
157        new_env.player_position = self.player_position
158        new_env.current_tile = self.current_tile
159        new_env.char_to_int = self.char_to_int
160        new_env.char_set = self.char_set
161        return new_env
162
163 def is_terminal(self):
164        return self.done"""
165
166        get_state_method = """def get_state(self):
167        return {"map": self.map}"""
168
169        step_method = """def step(self, action):
170        reward = 0
171        self.done = False
172        if action < 4:   # Movement actions
173            self.move_player(action)
174        self.done = reward > 0
175        info = {}
176        return self.get_state()["map"], reward, self.done, False, info
177        """
```

The following are the prompts used to edit the methods:

1. The state, render, and step method prompts are identical, except that the input function is replaced by the function currently in focus:

```
1        "Given the following get_state method of the class:\n" +
         get_state_method + "\n And the following game mechanic:\n" +
         mechanics + "\n Edit get_state method, if required, for the
         given game mechanic to work. Do not repeat the game mechanic as
          a method. Only output whole of the edited get_state method and
```

```
        if not edited, just output 'False'. Do not output anything
        else."
```

2. Adding helper methods:

```
    "Given the following methods: init method of the class:\n" +
    init_method + "\n All functions already present in the class:\n
    "+ other_methods + "\n The following game mechanic you just
    created:\n" + mechanics + "\n Add any new helper methods needed
     for the " + extract_function_name(mechanics) +" to work, if
    required. Do not create or update __init__, reset, step,
    get_state, render, get_action_space, or get_mechanics_to_action
     methods. Do not repeat the game mechanic as a method or the
    present methods. Do not add any new variables. Only output the
    additional new method or methods required, and if not added,
    just output 'False'. Do not output anything else."
```

3. The prompt to add new variables. `game\_mech\_class` one string with the whole initial class present in it.

```
    "Given the following class:\n" + game_mech_class + "\n And the
     following game mechanic:\n" + mechanics + "\n Add any new
    variables required in the GameMechEnv for the given game
    mechanic to work. Always add to the existing code. Only output
    the new variables if they are added in a Python dictionary
    format and if not added, just output 'False'. The Python
    dictionary can look like {'var_name_1': init_value, 'var_name_2
    ': init_value}. The keys should be the string of the variable
    name and the values should be the initial values. The values
    can never be a new argument to the init method. Do not output
    anything else."
```

4. To generate the action space

```
    "Given the following class:\n" + game_mech_class + "\n Also,
    the following game mechanic:\n" + mechanics + "\n And the
    following get_action_space method:\n" + action_space + "\n .
    Edit the get_action_space method to add more actions, if
    required, for the given game mechanic to work. Do not repeat
    the game mechanic as a method. Only output the edited
    get_action_space method and if not edited, just output 'False'.
     Do not output anything else."
```

5. To get a mapping of mechanics to actions.

```
    "Given the following class:\n" + game_mech_class + "\n Also,
    the following game mechanic:\n" + mechanics + "\n And the
    following get_mechanics_to_action method:\n" + mech_to_action +
     "\n . Edit the get_mechanics_to_action method to add more
    actions, if required, for the given game mechanic to work. The
    edition should be "+ extract_function_name(mechanics) +":
    action_number. Do not repeat the game mechanic as a method.
    Only output the edited get_mechanics_to_action method and if
    not edited, just output 'False'. Do not output anything else."
```

6. Game mechanics are then tested on a static environment:

```
1 str_world = """BBBBBBBBBB
2 BAAAAAAAAB
3 BAAAOAAAAB
4 BA#@OAAAAB
5 BA#AAAAAAB
6 BBBBBBBBBB"""
7
8 str_map_wo_chars = """BBBBBBBBBB
9 BAAAAAAAAB
```

```
10 BAAOOAAAAAB
11 BAAAOAAAAAB
12 BAAAAAAAAAB
13 BBBBBBBBBBB"""
14
15 walkables = ['A', 'B']
16 interactive_object_tiles = ['O']
17 enemy_tiles = ["#"]
18 npc_tiles = ["&"]
19 env_image = dict()
20
21
22 env_image["A"] = Image.open(r"world_tileset_data/
       td_world_floor_grass_c.png").convert("RGBA")
23 env_image["B"] = Image.open(r" world_tileset_data/
       td_world_wall_stone_h_a.png").convert("RGBA")
24 env_image["C"] = Image.open(r"world_tileset_data/
       td_world_floor_grass_c.png").convert("RGBA")
25 env_image["O"] = Image.open(r"world_tileset_data/td_world_chest.
       png").convert("RGBA")
26 env_image["@"] = Image.open(r"character_sprite_data/
       td_monsters_archer_d1.png").convert("RGBA")
27 env_image["#"] = Image.open(r"character_sprite_data/
       td_monsters_witch_d1.png").convert("RGBA")
28 env_image["&"] = Image.open(r"character_sprite_data/
       td_monsters_goblin_captain_d1.png").convert("RGBA")
29
30 env = GameMechEnv(walkable_tiles=walkables,
31      tiles_without_char=str_map_wo_chars,
32      tiles=env_image,
33      str_map_without_chars=str_map_wo_chars,
34      str_map=str_world,
35      interactive_object_tiles=interactive_object_tiles,
36      enemy_tiles=enemy_tiles)
```

## A.2   GAME GENERATION PROMPTS

All of the above prompts are also used in the game-generation pipeline. The difference is that we
keep track of the previously edited method so that, when the evaluation MCTS tree expands to the
next node, the prompt includes methods inherited from earlier nodes. This is necessary because each
newly expanded node introduces a new mechanic in addition to all prior mechanics.

After these generations, we generate a `make\_game` function. We start with generating a `env\`
`_dict\_func`, a function that returns a dictionary mapping tiles to their corresponding path func-
tions. We provide the following paths:

```
1  paths_to_tiles = r'''world_tileset_data/td_items_amulet_gold.png,
2      world_tileset_data/td_items_gem_ruby.png,
3      world_tileset_data/td_world_crate.png,
4      world_tileset_data/tg_world_barrel.png,
5      world_tileset_data/tg_world_floor_carpet_d.png,
6      world_tileset_data/tg_world_floor_moss_e.png,
7      world_tileset_data/tg_world_floor_sand_f.png,
8      world_tileset_data/tg_world_floor_panel_steel_c.png,
9      character_sprite_data/td_monsters_angel_d2.png,
10     character_sprite_data/td_monsters_archer_u2.png,
11     character_sprite_data/td_monsters_berserker_d1.png,
12     character_sprite_data/td_monsters_demon_l1.png'''
```

And the whole initial `env\_dict\_func`:

```
1  env_dict_func = '''def env_dict():
2  env_image = dict()
3  image_paths = dict()  # New dictionary to store paths
```

```
4
5      # Define a function to load image and store path
6      def load_image(char, path):
7          env_image[char] = Image.open(path).convert("RGBA")
8          image_paths[char] = path  # Store the path
9
10     # Load all images
11     base_path = r"/mnt/lustre/users/mnasir/gmd"
12     load_image("A", f"{base_path}/world_tileset_data/
       td_world_floor_grass_c.png")
13     load_image("B", f"{base_path}/world_tileset_data/
       td_world_wall_stone_h_a.png")
14     load_image("X", f"{base_path}/world_tileset_data/
       td_world_floor_grass_c.png")
15     load_image("O", f"{base_path}/world_tileset_data/td_world_chest.png")
16     load_image("I", f"{base_path}/world_tileset_data/td_world_chest.png")
17     load_image("C", f"{base_path}/world_tileset_data/td_world_chest.png")
18     load_image("@", f"{base_path}/character_sprite_data/
       td_monsters_archer_d1.png")
19     load_image("#", f"{base_path}/character_sprite_data/
       td_monsters_witch_d1.png")
20     load_image("&", f"{base_path}/character_sprite_data/
       td_monsters_goblin_captain_d1.png")
21
22     # Here you add any other tiles that are needed for the game in the
       same format
23
24     return env_image, image_paths '''
```

Therefore, the prompt for the env\_dict\_func generation:

```
1      "Given the game mechanics:\n" + mechanics[0] + "\nChange the
       following env_dict function to cater for the mechanics, if required
       .:\n" + env_dict_func + " Use the same paths for the same type of
       tiles that are being added in env_image, or use the following paths:\
       n"+ paths_to_tiles + "\n You don't have to use the paths provided if
       not needed. Strictly use the same paths. Do not use any other paths.
       Only add in env_image if needed. Return the full env_dict function."
```

Then we create a 2D character map through the prompt:

```
1      "Given the game mechanics:\n" + mechanics[0] + "\n The env_dict
       function, which has the paths for the tiles:\n" +
       env_dict_func_changed['choices'][0]['message']['content'] + "\nChange
        the following str_world in the str_map function to cater for the
       mechanics, if required.:\n" + str_map_func + "\n 'str_world' is the
       string that represents the 2D game map. Change it if only needed. It
       must always have 1 and only 1 '@' character, which represents the
       player. Return the full str_map function."
```

where the str\_map\_func is:

```
1      str_map_func = '''def str_map():
2
3          str_world = """AAAAAAAAAAAAAAAAAA
4      AAAAAAAAAAAAAAAAAA
5      AAA@OAXAAAAAAAAAAA
6      AAAAAAICAAAAAAAAAA
7      AAA#AAAAAAAAAAAAAA
8      AAAAAAAAAAAAAAAAAA"""
9
10         return str_world'''
```

Lastly, in the make\_game function we generate the important\_tiles\_func through the prompt:

```
"Given the game mechanics:\n" + mechanics[0] + "\n And the env_dict
function, which has the paths for the tiles:\n" +
env_dict_func_changed['choices'][0]['message']['content'] + "\n And
the str_map function, which has the string representation of the game
 map:\n" + str_map_func_changed['choices'][0]['message']['content'] +
 "\n Change the following important_tiles function to cater for the
mechanics, if required.:\n" + important_tiles_func + " Return the
full important_tiles function. Return all the tile types mentioned in
 the return statement of the function. Return empty list if the tile
type is not needed."
```

where the initial `important\_tiles\_func` is:

```
important_tiles_func = '''def important_tiles():
    walkables = ['A']   # Walkable tiles
    non_walkables = ['B']   # Non-walkable tiles
    interactive_object_tiles = ['O', 'I', 'C']   # Interactive objects
(e.g., chests)
    collectible_tiles = []   # Can add collectible tiles if needed
    npc_tiles = []   # Assume there are no NPCs represented in the
current setup
    player_tile = ['@']   # Player tile
    enemy_tiles = ['#', '&']   # Enemy tiles
    extra_tiles = [] # any other type of tiles for the game goes here
    return walkables, non_walkables, interactive_object_tiles,
collectible_tiles, npc_tiles, player_tile, enemy_tiles, extra_tiles'''
'
```

For game-mechanic generation, the terminal method is fixed, since we test each mechanic in isolation to verify that the MCTS agent can reach the end. In the subsequent game-generation stage, however, the terminal method may vary, as the win condition can change substantially.

```
"Given the game mechanics:\n" + get_games_scores.latest_methods['
mechanic'] + "\n" + mechanics[0] + "\n and the init function:\n" +
get_games_scores.latest_methods['init'] + "\n We want to train an
agent to play a game that uses these mechanics. The layout of the
game is the str_world in the following function:\n"+ get_games_scores
.latest_methods['str_world'] +"\n and the following function
describes what the tiles mean:\n"+get_games_scores.latest_methods['
tiles']+"\nThe following line describes the situation of the win
condition of the game:\n"+ "'"+line_response['choices'][0]['message'
]['content']+"'" +"\nWrite one method of the class which which wraps
win condition in it and tells the agent when the game will end. It
must focus on the mechanics in the game provided to you. All the
mechanics should be used to fulfill the win condition. The name of
the method should be is_terminal. Method should only return one
boolean variable. Only return the method and nothing else."
```

Here a `line\_response` is a win condition generated through the following prompt:

```
"Given the game mechanics:\n" + get_games_scores.latest_methods['
mechanic'] + "\n" + mechanics[0] + "\n write one line that describes
the win condition for the game that will use these mechanics. "
```

The name of the game is generated through:

```
"Given the game mechanics:\n" + get_games_scores.latest_methods['
mechanic'] + "\n" + mechanics[0] + "\nThe win condition for the game
in is_terminal method:\n " + is_terminal_response['choices'][0]['
message']['content'] + "\n And the line that explains the win
condition:\n " + line_response['choices'][0]['message']['content'] +
"\n Give the game a short name that describes the game well. Only
strictly output the name and nothing else. Should not have any
special characters in the name. Do not highlight the name."
```

## B  GENERATED GAME MECHANICS

### B.1  MECHANICS IN FIGURE 6

Following is the mechanic and helper functions for the game in Figure 6:

```
def spawn_unit(self):
    """Spawn a unit at an adjacent empty position"""
    reward = 0

    if len(self.units) >= self.max_units:
        return 0  # No penalty, just no reward

    player_row, player_col = self.player_position
    adjacency_offsets = [(0, -1), (0, 1), (-1, 0), (1, 0)]  #Left, Right,
     Up, Down

    # Try to find an empty adjacent position
    for dx, dy in adjacency_offsets:
        new_row = player_row + dx
        new_col = player_col + dy
        if (0 <= new_row < len(self.map) and 0 <= new_col <len(self.map
[0])):
            # Check the base tile type (without characters)
            base_tile = self.map_without_chars[new_row][new_col]
            current_tile = self.map[new_row][new_col]

            # Check if position is suitable for unit spawning
            if (base_tile in self.walkable_tiles and
                (new_row, new_col) not in self.units and
                (new_row, new_col) != self.player_position and
                current_tile not in self.enemy_tiles and
                current_tile in self.walkable_tiles):  # Current tile
should also be walkable

                # Spawn unit here
                unit_pos = (new_row, new_col)
                self.units.append(unit_pos)
                self.unit_health[unit_pos] = 100  # Initialize unit
health
                self.map[new_row][new_col] = self.unit_symbol
                reward = 1  # Reward for successful spawning
                break

    return reward

#------------------------

    def heal_unit(self):
        """Heal the selected unit"""
        reward = 0

        if not self.units or self.selected_unit >= len(self.units):
            return -1  # Penalty for invalid unit selection

        unit_pos = self.units[self.selected_unit]

        if unit_pos in self.unit_health:
            old_health = self.unit_health[unit_pos]
            self.unit_health[unit_pos] = min(100, self.unit_health[
    unit_pos] + 30)  # Heal 30 HP, max 100

            if old_health < self.unit_health[unit_pos]:
                heal_amount = self.unit_health[unit_pos] - old_health
```

```
 54                    print(f"Unit at {unit_pos} healed for {heal_amount} HP!
       Health: {self.unit_health[unit_pos]}")
 55                    reward = 1  # Small reward for healing
 56                else:
 57                    print(f"Unit at {unit_pos} is already at full health!")
 58                    reward = -1  # Penalty for unnecessary healing
 59
 60            return reward
 61
 62 #-------------------------
 63
 64 def player_attack(self):
 65     """Execute primary attack on adjacent targets"""
 66     reward = 0
 67     player_row, player_col = self.player_position
 68     adjacency_offsets = [(0, -1), (0, 1), (-1, 0), (1, 0)]
 69
 70     enemies_attacked = 0
 71     for dx, dy in adjacency_offsets:
 72         attack_row = player_row + dx
 73         attack_col = player_col + dy
 74         attack_pos = (attack_row, attack_col)
 75
 76         # Find enemy at this position
 77         for enemy in self.enemies:
 78             if enemy['pos'] == attack_pos:
 79                 damage = 25  # Player damage
 80                 enemy['health'] -= damage
 81                 enemies_attacked += 1
 82                 print(f"Player attacks {enemy['type']} for {damage}
       damage! Enemy health: {enemy['health']}")
 83
 84                 if enemy['health'] <= 0:
 85                     print(f"{enemy['type']} defeated!")
 86                     reward += 10  # Reward for defeating enemy
 87                 else:
 88                     reward += 2  # Small reward for successful attack
 89
 90     # Small penalty if no enemies to attack
 91     if enemies_attacked == 0:
 92         reward = -1
 93
 94     return reward
 95
 96 #-------------------------
 97
 98 def move_enemy_toward_target(self, enemy, target_pos):
 99     """Move enemy one step toward target"""
100     reward = 0
101     enemy_pos = enemy['pos']
102     enemy_row, enemy_col = enemy_pos
103     target_row, target_col = target_pos
104
105     # Calculate direction to move
106     row_diff = target_row - enemy_row
107     col_diff = target_col - enemy_col
108
109     # Choose move direction (simple pathfinding)
110     move_row, move_col = 0, 0
111     if abs(row_diff) > abs(col_diff):
112         move_row = 1 if row_diff > 0 else -1
113     else:
114         move_col = 1 if col_diff > 0 else -1
115
116     new_row = enemy_row + move_row
```

```
117       new_col = enemy_col + move_col
118
119       # Check if move is valid
120       if self._is_valid_enemy_move(enemy_pos, (new_row, new_col)):
121           self._execute_enemy_move(enemy, (new_row, new_col))
122
123       return reward
124
125  #-------------------------
126
127   def confuse_and_teleport_enemies(self):
128       """Apply area effect that disrupts enemy positioning"""
129       reward = 0
130       # Identify all enemy positions and create a list of positions
131       enemy_positions = []
132       for row in range(len(self.map)):
133           for col in range(len(self.map[0])):
134               if self.map[row][col] in self.enemy_tiles:  # Useactual enemy
         tiles from the map
135                   enemy_positions.append((row, col))
136       # If there are enemies on the map, confuse and possibly teleport them
137       if enemy_positions:
138           enemies_confused = 0
139           for enemy_row, enemy_col in enemy_positions:
140               # Randomly choose a direction to confuse the enemy
141               direction = random.choice(['up', 'down', 'left', 'right'])
142               teleport_possible = False
143               # Determine the new position for confusion
144               new_enemy_row, new_enemy_col = enemy_row, enemy_col
145               if direction == 'up' and enemy_row > 0:
146                   new_enemy_row -= 1
147               elif direction == 'down' and enemy_row < len(self.map) - 1:
148                   new_enemy_row += 1
149               elif direction == 'left' and enemy_col > 0:
150                   new_enemy_col -= 1
151               elif direction == 'right' and enemy_col < len(self.map[0]) -
         1:
152                   new_enemy_col += 1
153               # Instead of actually moving enemies, just count confusion
         attempts
154               distance_to_player = abs(enemy_row -self.player_position[0])
         + abs(enemy_col -self.player_position[1])
155               if distance_to_player <= 2:  # If enemy is within close range
156                   enemies_confused += 1
157
158           # Only give reward if multiple enemies were confused
159           if enemies_confused >= 2:
160               reward = 1
161       return reward
162
163  #-------------------------
164
165  def activate_and_combine_resources(self):
166       """Activates resource gathering and environmental interactionability.
         """
167       reward = 0
168       adjacency_offsets = [(0, -1), (0, 1), (-1, 0), (1, 0)]  # Up, Down,
         Left, Right
169       resource_tiles = ['R', 'F', 'W']  # R = Rock, F = Food, W = Wood
170
171       # Count adjacent resources and interactive objects
172       adjacent_resources = 0
173       adjacent_objects = 0
174
175       # Check for adjacent resources
```

```
176     for dx, dy in adjacency_offsets:
177         new_row = self.player_position[0] + dx
178         new_col = self.player_position[1] + dy
179         if 0 <= new_row < len(self.map) and 0 <= new_col <len(self.map
        [0]):
180             adjacent_tile = self.map[new_row][new_col]
181             if adjacent_tile in resource_tiles:
182                 adjacent_resources += 1
183
184     # Check for interactive objects nearby
185     for dx, dy in adjacency_offsets:
186         new_row = self.player_position[0] + dx
187         new_col = self.player_position[1] + dy
188         if 0 <= new_row < len(self.map) and 0 <= new_col <len(self.map
        [0]):
189             adjacent_tile = self.map[new_row][new_col]
190             if adjacent_tile in self.interactive_object_tiles:
191                 adjacent_objects += 1
192
193     # Only give reward for significant resource/object combinations
194     if adjacent_resources >= 2 and adjacent_objects >= 1:
195         reward = 5  # Reward only for optimal positioning
196
197     return reward
```

## B.2   MECHANICS IN FIGURE 7

The following is the mechanic and helper functions for the game in Figure 7:

```
1  def strategic_enemy_movement(self):
2      """Process all enemy actions for this turn using A*path finding"""
3      import heapq
4      reward = 0
5
6      def heuristic(pos, goal):
7          return abs(pos[0] - goal[0]) + abs(pos[1] - goal[1])
8
9      for enemy in self.enemies[:]:
10         if enemy['finished']:
11             continue
12
13         if self.turn_counter - enemy['last_move_turn'] >= self.
       enemy_move_cooldown:
14             enemy['last_move_turn'] = self.turn_counter
15
16             # A* pathfinding to find next best move
17             start = enemy['pos']
18             goal = self.goal_position
19
20             # Priority queue: (f_score, g_score, position, path)
21             open_set = [(heuristic(start, goal), 0, start,[start])]
22             closed_set = set()
23
24             directions = [(-1, 0), (1, 0), (0, -1), (0, 1)]
25             max_iterations = 50  # Limit search to prevent lag
26             iterations = 0
27             optimal_move = None
28
29             while open_set and iterations < max_iterations:
30                 iterations += 1
31                 f_score, g_score, current, path =heapq.heappop(open_set)
32
33                 if current in closed_set:
34                     continue
```

```
35
36                  if current == goal:
37                      # Return the next move in the optimal path
38                      optimal_move = path[1] if len(path) > 1 elseNone
39                      break
40
41                  closed_set.add(current)
42
43                  for dx, dy in directions:
44                      neighbor = (current[0] + dx, current[1] + dy)
45
46                      if (neighbor in closed_set or
47                          notself._is_valid_position_for_pathfinding(
    neighbor)):
48                          continue
49
50                      new_g_score = g_score + 1
51                      new_f_score = new_g_score +heuristic(neighbor, goal)
52                      new_path = path + [neighbor]
53
54                      heapq.heappush(open_set, (new_f_score,new_g_score,
    neighbor, new_path))
55
56          # Execute move if valid
57          if optimal_move and self._is_valid_enemy_move(enemy['pos'],
    optimal_move):
58              self._execute_enemy_move(enemy, optimal_move)
59
60          # Check if enemy reached goal
61          if enemy['pos'] == self.goal_position:
62              if not enemy['finished'] and notself.game_finished:
63                  enemy['finished'] = True
64                  self.game_finished = True
65              self.completion_order.append(f"enemy_{enemy['type']}")
66                  print(f"Enemy {enemy['type']} reached thegoal first!
    ENEMY WINS!")
67                  reward -= 100  # Player loses big when enemy wins
68
69      return reward
70
71  #------------------------
72
73  def _is_valid_position_for_pathfinding(self, pos):
74      """Check if position is valid for pathfinding (allows temporary
        occupation)"""
75      row, col = pos
76      if not (0 <= row < len(self.map) and 0 <= col < len(self.map[0])):
77          return False
78
79      tile = self.map[row][col]
80      # Allow movement through walkable tiles and goal
81      return tile in self.walkable_tiles or tile == 'F'
82
83  #------------------------
84
85  def _is_valid_enemy_move(self, current_pos, new_pos):
86      """Check if enemy move is valid"""
87      new_row, new_col = new_pos
88      if not (0 <= new_row < len(self.map) and 0 <= new_col < len(self.
        map[0])):
89          return False
90
91
92      current_tile = self.map[new_row][new_col]
93
94      # Can move to walkable tiles or flag
```

```
95      if current_tile not in self.walkable_tiles and current_tile != 'F':
96          return False
97
98      # Cannot move to position occupied by player or other enemies
99      if new_pos == self.player_position:
100         return False
101
102     for other_enemy in self.enemies:
103         if other_enemy['pos'] == new_pos:
104             return False
105
106     return True
```

## C QUALITY-DIVERSITY

### C.1 INITIAL MECHANICS

Here we will mention the aspects of the quality-diversity (QD) algorithm that would help in reproducibility, and were not mentioned in the main paper. The following are the initial mechanics used to initialise the QD algorithm:

```
1   mech_1 = """\ndef move_player(self, action):
2   moves = {0: (-1, 0), 1: (1, 0), 2: (0, -1), 3: (0, 1)}  # Up, Down,
    Left, Right
3   dx, dy = moves[action]
4   new_row = self.player_position[0] + dx
5   new_col = self.player_position[1] + dy
6   reward = 0
7   if 0 <= new_row < len(self.map) and 0 <= new_col < len(self.map[0]):
8       new_tile = self.map[new_row][new_col]
9       if new_tile in self.walkable_tiles:
10          self.update_player_position(new_row, new_col, new_tile)
11  return reward"""
12
13  #-------------------------------
14
15  mech_2 = """\ndef pick_object(self):
16      reward = 0
17      # Check adjacent tiles for interactive objects and pick them if
        present
18      adjacent_positions = [(0, -1), (0, 1), (-1, 0), (1, 0)]  # Up, Down,
        Left, Right
19      for dx, dy in adjacent_positions:
20          row, col = self.player_position  # player_position is in (row,
        col) format
21          new_row = row + dx
22          new_col = col + dy
23          if 0 <= new_row < len(self.map) and 0 <= new_col < len(self.map
        [0]):
24              target_tile = self.map[new_row][new_col]
25              if target_tile in self.interactive_object_tiles:
26                  self.map[new_row][new_col] = self.default_walkable_tile
27                  reward = 1
28                  break  # Exit after picking up one object
29      return reward"""
30
31  #-------------------------------
32
33
34  mech_3 = """\ndef hit_enemy(self):
35      reward = 0
36      # Check adjacent tiles for enemies and hit them if present
```

```
37      adjacent_positions = [(0, -1), (0, 1), (-1, 0), (1, 0)]  # Up, Down,
        Left, Right
38      for dx, dy in adjacent_positions:
39          row, col = self.player_position  # player_position is in (row,
        col) format
40          new_row = row + dx
41          new_col = col + dy
42          if 0 <= new_row < len(self.map) and 0 <= new_col < len(self.map
        [0]):   # Check grid bounds
43              target_tile = self.map[new_row][new_col]
44              if target_tile in self.enemy_tiles:
45                  self.map[new_row][new_col] = self.default_walkable_tile
46                  reward = 1
47                  break  # Exit after hitting one enemy
48      return reward"""
49
50  #-------------------------------
51
52  mech_4 = """\ndef teleport_player(self):
53      # Find all walkable tiles that are not adjacent to the player
54      non_adjacent_walkable_positions = []
55      adjacency_offsets = [(0, -1), (0, 1), (-1, 0), (1, 0)]  # Up, Down,
        Left, Right
56      reward = 0
57      # Search the map for walkable and non-adjacent tiles
58      for row in range(len(self.map)):
59          for col in range(len(self.map[0])):
60              if self.map[row][col] in self.walkable_tiles:
61                  is_adjacent = False
62                  for dx, dy in adjacency_offsets:
63                      if (row == self.player_position[0] + dx) and (col ==
        self.player_position[1] + dy):
64                          is_adjacent = True
65                          break
66                  if not is_adjacent:
67                      non_adjacent_walkable_positions.append((row, col))
68      # Teleport the player to a random walkable, non-adjacent position
69      if non_adjacent_walkable_positions:
70          new_position = random.choice(non_adjacent_walkable_positions)
71          self.update_player_position(new_position[0], new_position[1],
        self.map[new_position[0]][new_position[1]])
72          reward += 1
73      return reward"""
74
75  #-------------------------------
76
77  mech_5 = """\ndef swap_positions(self):
78      # Find all enemy positions on the map
79      enemy_positions = []
80      reward = 0
81      for row in range(len(self.map)):
82          for col in range(len(self.map[0])):
83              if self.map[row][col] in self.enemy_tiles:
84                  enemy_positions.append((row, col))
85      # If there are enemies, randomly swap the player's position with an
        enemy's position
86      if enemy_positions:
87          swap_with = random.choice(enemy_positions)
88          enemy_row, enemy_col = swap_with
89          player_row, player_col = self.player_position
90          # Swap positions of player and enemy on the map
91          self.map[player_row][player_col], self.map[enemy_row][enemy_col]
        = self.map[enemy_row][enemy_col], self.map[player_row][player_col]
92          # Update the player's position to the swapped position
93          self.player_position = (enemy_row, enemy_col)
```

```
 94          # Optional: Output the result of the swap
 95          reward += 1
 96      return reward"""
 97
 98  #-------------------------------
 99
100  mech_6 = """\ndef push_object(self):
101      reward = 0
102      adjacent_positions = [(0, -1), (0, 1), (-1, 0), (1, 0)]  # Up, Down,
     Left, Right
103      for dy, dx in adjacent_positions:  # Swapped to dy, dx to match map
     indexing
104          y, x = self.player_position  # Player position is in (row, col)
     format
105          new_y, new_x = y + dy, x + dx
106          if 0 <= new_y < len(self.map) and 0 <= new_x < len(self.map[0]):
      # Check bounds
107              target_tile = self.map[new_y][new_x]
108              if target_tile in self.interactive_object_tiles:
109                  push_y, push_x = new_y + dy, new_x + dx  # Push in same
     direction
110                  if 0 <= push_y < len(self.map) and 0 <= push_x < len(self
     .map[0]):
111                      if self.map[push_y][push_x] in self.walkable_tiles:
112                          self.map[push_y][push_x] = target_tile
113                          self.map[new_y][new_x] = self.
     default_walkable_tile
114                          reward = 1
115                          break
116      return reward"""
117
118  #-------------------------------
119
120  mech_7 = """\ndef jump_player(self):
121      reward = 0
122      # Define possible jump directions
123      jump_directions = [(0, -2), (0, 2), (-2, 0), (2, 0)]  # Up, Down,
     Left, Right (2 tiles)
124      for dx, dy in jump_directions:
125          row, col = self.player_position  # player_position is in (row,
     col) format
126          mid_row, mid_col = row + dx // 2, col + dy // 2  # Middle tile (
     jumped over)
127          new_row, new_col = row + dx, col + dy  # Landing tile
128          # Check if the jump is within bounds
129          if 0 <= new_row < len(self.map) and 0 <= new_col < len(self.map
     [0]):
130              target_tile = self.map[new_row][new_col]
131              # Check if the landing tile is walkable
132              if target_tile in self.walkable_tiles:
133                  # Perform the jump
134                  self.update_player_position(new_row, new_col, target_tile
     )
135                  reward = 1
136                  break  # Exit after a successful jump
137      return reward"""
138
139  #-------------------------------
140
141  mech_8 = """\ndef drop_object(self):
142      reward = 0
143      # Check adjacent tiles for empty walkable space
144      adjacent_positions = [(0, -1), (0, 1), (-1, 0), (1, 0)]  # Up, Down,
     Left, Right
145      for dx, dy in adjacent_positions:
```

```
146            row, col = self.player_position
147            new_row = row + dx
148            new_col = col + dy
149            # Check if position is within bounds and walkable
150            if 0 <= new_row < len(self.map) and 0 <= new_col < len(self.map
       [0]):
151                if self.map[new_row][new_col] in self.walkable_tiles:
152                    # Place an interactive object
153                    self.map[new_row][new_col] = self.
       interactive_object_tiles[0]  # Using first interactive object tile
154                    reward = 1
155                    break  # Exit after dropping one object
156        return reward"""
157
158 mech_9 = """\ndef enemy_move(self):
159        reward = 0
160        # Find all enemy positions with "#" tile on the map
161        enemy_positions = []
162        for row in range(len(self.map)):
163            for col in range(len(self.map[0])):
164                if self.map[row][col] == "#":
165                    enemy_positions.append((row, col))
166
167        # If there are enemies, move one randomly
168        if enemy_positions:
169            # Pick a random enemy to move
170            enemy_row, enemy_col = random.choice(enemy_positions)
171
172            # Define possible move directions (same as player)
173            moves = {0: (-1, 0), 1: (1, 0), 2: (0, -1), 3: (0, 1)}  # Up,
       Down, Left, Right
174
175            # Try each direction randomly until we find a valid move
176            directions = list(moves.keys())
177            random.shuffle(directions)
178
179            for action in directions:
180                dx, dy = moves[action]
181                new_row = enemy_row + dx
182                new_col = enemy_col + dy
183
184                # Check if the new position is valid
185                if 0 <= new_row < len(self.map) and 0 <= new_col < len(self.
       map[0]):
186                    new_tile = self.map[new_row][new_col]
187                    if new_tile in self.walkable_tiles:
188                        # Move the enemy
189                        self.map[enemy_row][enemy_col] = self.
       default_walkable_tile
190                        self.map[new_row][new_col] = "#"
191                        break  # Exit after successful move
192
193        return reward"""
194
195 mech_10 = """\ndef enemy_hit(self):
196        reward = 0
197        # Find all enemy positions with "#" tile on the map
198        enemy_positions = []
199        for row in range(len(self.map)):
200            for col in range(len(self.map[0])):
201                if self.map[row][col] == "#":
202                    enemy_positions.append((row, col))
203
204        # Check if any enemy is adjacent to the player and can hit
205        player_row, player_col = self.player_position
```

```
206    adjacent_positions = [(0, -1), (0, 1), (-1, 0), (1, 0)]  # Up, Down,
       Left, Right
207
208    for enemy_row, enemy_col in enemy_positions:
209        # Check if this enemy is adjacent to the player
210        for dx, dy in adjacent_positions:
211            check_row = enemy_row + dx
212            check_col = enemy_col + dy
213            # If the adjacent position matches the player's position
214            if check_row == player_row and check_col == player_col:
215                # Enemy hits the player
216                reward = -1  # Negative reward for player getting hit
217                break  # Exit after first hit (one enemy hitting is
       enough)
218        if reward != 0:  # If a hit occurred, stop checking other enemies
219            break
220    return reward"""
```

## C.2 GAME MECHANICS TYPES

We specify the types of mechanics that MORTAR uses to compute similarity scores for the Quality–Diversity archive. For each mechanic, we list its category followed by the keywords used to determine similarity.

- **Movement**: move, walk, run, jump, fly, teleport, dash, swim, climb, crouch, sprint

- **Interaction**: pick, use, interact, open, close, talk, trade, craft, activate, push, pull

- **Combat**: attack, fight, hit, shoot, defend, block, dodge, cast, spell, heal, damage

- **Progression**: level, upgrade, unlock, improve, evolve, progress, achieve, complete, quest, mission

- **Environment**: weather, day, night, season, climate, destroy, build, terraform, grow, plant

- **Puzzle**: solve, puzzle, riddle, match, connect, arrange, decode, decipher, logic, pattern

- **Resource Management**: collect, gather, manage, inventory, store, spend, earn, balance, allocate, distribute

- **Exploration**: explore, discover, map, reveal, uncover, navigate, search, investigate, scout, survey

- **Time Manipulation**: time, slow, fast, rewind, forward, pause, resume, loop, cycle, sequence

## C.3 PROMPTS FOR EVOLUTIONARY OPERATORS

The following are the prompts for the evolutionaru operators:

1. **Mutation:**

```
1    "Create a new game mechanic from the given mechanic that
     extends its features:\n" + solution[0] + "\n Do not make any
     assumptions, if you want to add a new variable or a new
     function, you should do it within the game mechanic method. The
      mechanic must return a reward, which is an integer. If a tile
     is being assumed then it should be defined as a single capital
     alphabet character and not a word. If a player is being assumed
      then it should be '@' tile. Remember that the game mechanic
     function should only take 'self' as parameter. Only output the
     new game mechanic as Python function, nothing else."
```

2. **Diversity Mutation:**

```
1      "Create a new game mechanic that is different, in terms of
       behavior of mechanics, from the ones provided:\n" + solution[0]
        + "\n Do not make any assumptions, if you want to add a new
       variable or a new funciton, you should do it within the game
       mechanic method. The mechanic must return a reward, which is an
        integer. If a tile is being assumed then it should be defined
       as a single capital alphabet character and not a word. If a
       player is being assumed then it should be '@' tile. Remember
       that the game mechanic function should only take 'self' as
       parameter. Only output the new game mechanic as Python function
       , nothing else."
```

3. **Compatibility Mutation:**

```
1      "Create a new game mechanic that will make the game better
       when combined with the following game mechanics:\n" + solution
       + "\n Do not make any assumptions, if you want to add a new
       variable or a new funciton, you should do it within the game
       mechanic method. The mechanic must return a reward, which is an
        integer. If a tile is being assumed then it should be defined
       as a single capital alphabet character and not a word. If a
       player is being assumed then it should be '@' tile. Remember
       that the game mechanic function should only take 'self' as
       parameter. The name of the mechanic should be coherent with the
        behaviour of it. Only output the new game mechanic as Python
       function, nothing else."
```

4. **Crossover:**

```
1      "Create a new game mechanic that combines the features of the
       given two mechanics to create a new game mechanic that combines
        the behavior of the both of them:\n" + solution + "\n Do not
       make any assumptions, if you want to add a new variable or a
       new method, you should do it within the function. The mechanic
       must return a reward, which is an integer. If a tile is being
       assumed then it should be defined as a single capital alphabet
       character and not a word. If a player is being assumed then it
       should be '@' tile. Remember that the game mechanic function
       should only take 'self' as parameter. The name of the mechanic
       should be coherent with the behaviour of it. Only output the
       new game mechanic as Python function, nothing else."
```

# D  GAMES

Play games in the user study by following the links:

1. TreasureHunt:https://mortar-x3p7.onrender.com/games/TreasureHunt

2. HeroBreakout:https://mortar-x3p7.onrender.com/games/HuntBreakout

3. AllyCraft:https://mortar-x3p7.onrender.com/games/AllyCraft

4. CrystalCavernsCommander:https://mortar-x3p7.onrender.com/games/Crystal_Caverns_Commander

5. MagneticProwess:https://mortar-x3p7.onrender.com/games/MagneticProwess

6. HeroHunt:https://mortar-x3p7.onrender.com/games/HeroHunt

