# OpenReview forum: "Mortar: Evolving Mechanics For Automatic Game Design"
_ICLR.cc/2026/Conference — Submitted to ICLR 2026_

### Official Review · Reviewer_pjAC · 2025-10-20

**Soundness:** 1
**Presentation:** 3
**Contribution:** 2
**Rating:** 2
**Confidence:** 4

**Summary:**

The paper addresses automated game generation by composing game mechanics, evaluating candidate games based on the ranking of agents of differing capabilities, and guiding search by estimating the marginal contribution of a mechanic to player rankings. The proposed method specifically uses MCTS to construct games by incrementally adding game mechanics. Mechanic values are estimated with a novel metric defined based on Shapley values to estimate the marginal contribution of each mechanic to the overall player ranking.

This approach is used in a quality-diversity algorithm that measures diversity through similarity of LLM-proposed mechanic names to a set of category types. Evolution operators are used to: (1) generate mechanics differing from prior examples, (2) combine mechanics where there is similar code structure, and (3) mutate mechanics.

Evaluation results compare the system to ablations of the search algorithm and also assess whether the total archive shows improvements in the new metric.

**Strengths:**

# originality
Estimating the marginal contribution of mechanics (derived from Shapley values) is the key new idea. This is a useful idea that would generalize across many search techniques should it prove useful as a signal for mechanic value.

# quality
Compares to ablations to establish a notion of baseline performance.

# clarity
Methodology is easy to follow and the results provide a narrative of the outcomes that are hard to qualitatively assess.

# significance
Of relevance to the game generation and game playing communities. The mechanic generation idea might be extensible to task generation in other domains (ex: robotics) with sufficient modifications.

**Weaknesses:**

# quality
Overall it is hard to tell from the experiments that there is substantial difference in outcomes among the methods. Further details are in the questions below.

No cost comparisons are made. It is hard to tell if MCTS has a greater budget and this is the primary driver of performance differences.

# clarity
See below for questions on specific metrics, tables, and interpretation.

**Questions:**

# questions
- lines 110-112: It seems like the relevant similarity would be the code to the description, not the mechanic name alone, as it can diverge arbitrarily in implementation. Or is there evidence the implementations tend to be very close?
- lines 244-245: Related to the above: is it clear that the implementations match the categories? What evidence is there?
- Is rank a good metric given two of the baselines would tend to be so bad as to inflate rankings? Ranks have no notion of relative distance among models, which makes it hard to tell how much the additional computational budgets of agents are differentiating performance.
- lines 261-262: How were costs (budgets) made equivalent among the methods?
	- For example: evaluating the same number of games during generation
- Table 1
	- Which differences are statistically significant and what are their effect sizes? From inspection the bounds look to overlap.
	- Is there any "diversity only" metric to use?
		- The ideal for generation is often is to produce a few, very good, very diverse outcomes. 100s of elites is a lot for humans to filter.
	- How to interpret number of elites?
		- Random Selection does very well. Is that the simply the number of archive cells populated?
		- The success rate of ~12% vs 17% (with 5 percentage point deviations) suggests something is very compressed here. Was the range of outcomes was too constrained by the QD setup? The game mechanic space? The lack of clear separation makes it unclear how much the method matters for this task design.
- Table 2
	- Same question as above about statistical testing. I'm not sure about a correlation: tau 0.4 vs 0.5 have total scores 3 and 22 respectively in the first block; in the third block tau 0.6 vs 0.3 also have scores 10 vs 16. The scatter plot would not immediately suggest an linear relationship.


# suggestions / minor comments
- Sokoban Initialisation: It would be nice to have this crossed with some other generators (perhaps only random) to establish a baseline to compare and assess the hypothesis about initialization sensitivity.
- lines 271-273: "pair them according to their distribution" - What distribution?
- line 343: "once players discover the optimal path, the game loses replay value"
	- Which agents have memorization to recognize replay value? Random would not, nor would Greedy on my understanding of the implementation.

---

### Official Review · Reviewer_pFKd · 2025-10-31

**Soundness:** 3
**Presentation:** 2
**Contribution:** 3
**Rating:** 6
**Confidence:** 4

**Summary:**

This work presents a system that leverages an LLM to evolve quality-diverse game mechanics. The LLM serves as an operator to generate offspring during the evolution process and defines the win condition. The quality is evaluated based on an estimation of how the game distinguishes players with different skill levels (simulated with MCTS using different parameters). This makes sense to me, but it would be better if the authors could find some references to support this choice. The diversity archive uses mechanic type and code complexity as the two dimensions. The authors also conducted a user study to evaluate the results. I played the online demo and found some of the generated games interesting.

**Strengths:**

Game generation is a challenging problem but has not gained enough attention. This paper demonstrates an attempt towards generating complete games purely through AI. It generates both game mechanics and level layouts simultaneously. While it does not create assets such as tile images, modern computer vision techniques should make generating such assets relatively straightforward. The methodology of using an LLM as an evolutionary operator is practical. The system is implemented based on Python rather than a domain-specific language, making it more general. Based on my research experience in procedural content generation, this paper represents a remarkable advance in this field. It also offers several promising research opportunities for future works, e.g., testing generalization of reinforcement learning agents.

**Weaknesses:**

The paper does not make its key contributions and technical content sufficiently clear. I would recommend that the authors explicitly summarize their contributions and novelty beyond closely related papers (e.g., Gavel (Todd et al., 2024)) in the introduction, even though I personally understand such contributions and novelty. It would be appreciated if the authors could provide some visualizations of how the LLM operates on the game in the appendix.

Meanwhile, the process of level layout generation remains unclear to me. Only Figure 1 mentions a string_map() function described as "generate 2D string map of the game," but it is not explained how this was implemented.

In the online demo, the control keys are quite confusing. I think it would be better to follow common conventions (e.g., using W, A, S, D for up, left, down, right). This should be easy to address through predefined rules or LLMs.

Figure 2 should have its axes labeled, though I understand from the context that they likely represent code complexity and mechanic type.

**Questions:**

1. How are the level layouts generated? The paper only mentions a string_map() function that generates a 2D string map, but the underlying implementation remains unclear.
2. I noticed a "story" in the online demo. How was this story generated? Was it created by the system, or was it predefined and provided to the LLM as a prompt?
3. How does the system select tile images?

---

### Official Review · Reviewer_zX6Y · 2025-11-01

**Soundness:** 2
**Presentation:** 3
**Contribution:** 2
**Rating:** 2
**Confidence:** 4

**Summary:**

The authors present an LLM-powered quality-diversity game generation system they call MORTAR. They cover the quality-diversity and code generation aspects of the system, along with a quality/fitness function adapted from Nielsen et al. They then present some results showing that  the system is able to effectively optimize this quality function. Finally they present the results of a small human subject study, which demonstrates no consistent relationship between their quality function and human evaluation.

**Strengths:**

The originality of the work primarily comes from generating games with python code, thus allowing for a larger search space than prior work in the area. This is reasonable originality as python certainly represents a larger search space than PuzzleScript or Ludii. The quality of the work is more mixed. The system is described well and the implementation follows nicely from prior work, particularly the reliance on Nielsen et al.'s evaluation approach. The novelty of trying to identify what mechanics most contributed to this quality function is also interesting. However, the results do not clearly support the claims in the paper or effectively evaluate the utility of its novel elements. In terms of clarity the paper is well-written, and the figures are largely clear. Finally, the significance is low. There are many game generation and mechanic generation papers, and many now involving LLMs.

**Weaknesses:**

The current draft of this paper has a number of weaknesses that could be improved.

First, the paper makes an odd claim around there being "comparatively little attention" paid to generating mechanics. This does not follow from the literature, or even from the related work section in this paper. More broadly there is an issue in the introduction of motivation. It's not clear why we need MORTAR, given the large amount of existing game generation approaches.

Second, and most importantly, the current evaluation and results do not interrogate the novel aspects of this work. Currently, many of the experiments and results (Figure 3 and Table 1) simply identify that the authors' approach can optimize the given quality function, and that the authors' approach outperforms some ablations when doing this. The user study similarly evaluates the comparative quality of some games, and the results do not support that Nielsen et al.'s evaluation framework aligns with human evaluation. This is itself a useful finding for the technical games research community, but it does not evaluate the novel aspects of this work.

The novel aspects of this work come from the CITS score and the use of python instead of a domain-specific language. Evaluations that evaluated these elements would have been much more useful. For example, the authors could have measured the extent to which the CITS core aligned with human annotators or other ways of determining mechanical importance. The authors also could have considered a comparison between MORTAR games and games from other game generators.

Outside of these issues with the experiments and results, Figure 1 and Figures 4 and 5 are both difficult to read. Figure 1 is dense with a large amount of small text. Figures 4 and 5 appear to have hand-drawn arrows, which are difficult to parse. A revision of these figures would be valuable to improve the clarity of the paper.

**Questions:**

1. Why focus on game quality evaluation, given the paper's technical contributions?

---

### Official Review · Reviewer_9vdu · 2025-11-01

**Soundness:** 3
**Presentation:** 3
**Contribution:** 3
**Rating:** 4
**Confidence:** 4

**Summary:**

This paper introduces MORTAR, a framework for evolving game mechanics using a combination of a quality-diversity (QD) evolutionary algorithm and a large language model LLM. Mechanics are represented as code snippets that can be combined into playable games, which are evaluated using skill gradient consistency as a proxy for game quality. The paper also proposes Constrained Importance Through Search (CITS) to estimate each mechanic’s contribution to game depth. Results include small-scale ablations and a user study showing that games with higher skill gradients tend to be preferred by players.

**Strengths:**

- Interesting and original focus on mechanic generation rather than level or asset generation, which is underexplored in procedural content generation.
- Creative use of LLM-guided mutation and code evolution within a structured search framework.
- Conceptually elegant link between “skill gradient” and perceived game depth.
- The paper is readable and technically ambitious, combining ideas from QD search, MCTS evaluation, and code-based generative design.

**Weaknesses:**

- Not sure if it is that relevant for ICLR, which is broadly about learning representations
- The claim that “a game’s quality is revealed through a consistent skill gradient” seems too strong. Many successful games (e.g., Animal Crossing, Cards Against Humanity, The Sims) are not skill-based yet still compelling.
- The user study (N=10) is rather small
-  It’s unclear how MORTAR compares to simpler or ablated variants without LLM-driven mutations. The relative importance of components could be analysed in more depth.
- Ideally there would be another baseline to compare the current approach to, e.g. human-designed mechanic set, grammar or template based mechanic generation

**Questions:**

1. How do you distinguish between generating new mechanics and entirely new games?
2. How sensitive are results to the choice of agents in the skill gradient evaluation?
3. How do you measure or validate that a mechanic is truly novel?

---

### Meta-Review · Area_Chair_gq3s · 2025-12-06

**Summary:**

The authors present an LLM-powered quality-diversity game generation system named MORTAR. Four reviewers thoroughly reviewed this paper, with 2 reject (Rating 2), 1 marginally below the acceptance threshold (Rating 4), and 1 marginally above the acceptance threshold (Rating 6). The reviewers raised some reasonable concerns, including:

Reviewer 9vdu

1.	The user study is too small.

2.	Comparisons with LLM-driven mutations and other baselines.

3.	Lack of analysis of components.

Reviewer zX6Y

1.	Motivation and novelty of MORTAR.

2.	The clarity of the paper.

Reviewer pFKd

1.	The contributions of MORTAR are unclear.

2.	The process of level layout generation remains unclear.

3.	The control keys are quite confusing.

Reviewer pjAC

1.	The budget of MCTS.

2.	The implementation and evaluation details.

3.	The statistical analysis.

**During the rebuttal period, the authors did not respond to the reviewers’ comments.** Based on the unsolved concerns and unclear contributions, I recommend rejecting this paper.

**Reviewer Concerns:**

During the rebuttal period, the authors did not respond to the reviewers’ comments. All of their concerns are not well addressed.

**Reviewer Scores:**

During the rebuttal period, the authors did not respond to the reviewers’ comments. I think all reviewers would not change their scores.

---

### Decision · Program_Chairs · 2026-01-26

Reject